# Overexpression of Long-Chain Acyl-CoA Synthetase 5 Increases Fatty Acid Oxidation and Free Radical Formation While Attenuating Insulin Signaling in Primary Human Skeletal Myotubes

**DOI:** 10.3390/ijerph16071157

**Published:** 2019-03-31

**Authors:** Hyo-Bum Kwak, Tracey L. Woodlief, Thomas D. Green, Julie H. Cox, Robert C. Hickner, P. Darrell Neufer, Ronald N. Cortright

**Affiliations:** 1Department of Physiology, East Carolina University, Greenville, NC 27858, USA; kwakhb@inha.ac.kr (H.-B.K.); greent@ecu.edu (T.D.G.); neuferp@ecu.edu (P.D.N.); 2Department of Kinesiology, East Carolina University, Greenville, NC 27858, USA; woodlieft18@ecu.edu (T.L.W.); coxju@ecu.edu (J.H.C.); 3Department of Kinesiology, Inha University, Incheon 22212, Korea; 4Department of Pharmacology and Toxicology, East Carolina University, Greenville, NC 27858, USA; 5Department of Internal Medicine, East Carolina University, Greenville, NC 27858, USA; 6The East Carolina Diabetes and Obesity Institute, East Carolina University, Greenville, NC 27858, USA; 7Department of Nutrition, Food and Exercise Sciences, Florida State University, Tallahassee, FL 32306, USA; rhickner@fsu.edu

**Keywords:** ACSL-5, fatty acid oxidation, insulin signaling, mitochondria, ROS, skeletal muscle

## Abstract

In rodent skeletal muscle, acyl-coenzyme A (CoA) synthetase 5 (ACSL-5) is suggested to localize to the mitochondria but its precise function in human skeletal muscle is unknown. The purpose of these studies was to define the role of ACSL-5 in mitochondrial fatty acid metabolism and the potential effects on insulin action in human skeletal muscle cells (HSKMC). Primary myoblasts isolated from vastus lateralis (obese women (body mass index (BMI) = 34.7 ± 3.1 kg/m^2^)) were transfected with ACSL-5 plasmid DNA or green fluorescent protein (GFP) vector (control), differentiated into myotubes, and harvested (7 days). HSKMC were assayed for complete and incomplete fatty acid oxidation ([1-^14^C] palmitate) or permeabilized to determine mitochondrial respiratory capacity (basal (non-ADP stimulated state 4), maximal uncoupled (carbonyl cyanide-4-(trifluoromethoxy)phenylhydrazone (FCCP)-linked) respiration, and free radical (superoxide) emitting potential). Protein levels of ACSL-5 were 2-fold higher in ACSL-5 overexpressed HSKMC. Both complete and incomplete fatty acid oxidation increased by 2-fold (*p* < 0.05). In permeabilized HSKMC, ACSL-5 overexpression significantly increased basal and maximal uncoupled respiration (*p* < 0.05). Unexpectedly, however, elevated ACSL-5 expression increased mitochondrial superoxide production (+30%), which was associated with a significant reduction (*p* < 0.05) in insulin-stimulated p-Akt and p-AS160 protein levels. We concluded that ACSL-5 in human skeletal muscle functions to increase mitochondrial fatty acid oxidation, but contrary to conventional wisdom, is associated with increased free radical production and reduced insulin signaling.

## 1. Introduction

Growing scientific evidence suggests that a central and precipitating factor for the development of obesity-associated skeletal muscle insulin resistance is the presence of a cellular lipid load that exceeds mitochondrial energy demand, leading to organelle dysfunction and disruption [1,2,3,4,5,6]. In support of this hypothesis, several reports have indicated that mitochondrial lipid oxidation is impaired in skeletal muscle from obese and type 2 diabetic subjects [7,8,9,10,11], likely due in part to reduced mitochondrial content and/or respiratory activity [8].

Long-chain acyl-CoA synthetase (ACSL) catalyzes the ligation of coenzyme A (Co-A) to long-chain fatty acids to form fatty acyl-CoA, which is then partitioned to either the mitochondria for oxidation or toward the triacylglycerol synthesis pathway for storage [12,13]. In addition to declines in mitochondrial content, skeletal muscle lipid oxidation could be reduced by impaired ACSL function leading to the accumulation of lipid intermediates including diacylglycerols, which are thought to activate serine/threonine kinases (e.g., inhibitor of nuclear factor kappa-B kinase subunit beta (IKKβ), c-Jun N-terminal kinase (JNK), protein kinase C theta (PKCθ)) [14] that phosphorylate and inhibit insulin receptor substrate 1 (IRS1), thus downregulating insulin signaling [3,6,15,16,17]. Total ACSL activity has been previously noted in skeletal muscle from obese women [18,19], yet isoform-specific function within human skeletal muscle has not been determined.

To date, five ACSL isoforms have been identified and characterized in rodents and humans [20,21,22,23]. Each of the ACSL family members is thought to have a distinct tissue distribution, intracellular localization (e.g., endoplasmic reticulum, mitochondria, peroxisomes), and expression level in response to metabolic challenges [12,13,22,24,25,26,27,28]. The majority of data on ACSL function has been obtained in liver from rodents with far fewer studies conducted in skeletal muscle [28], and even fewer from humans. In addition, whereas several ACSL isoforms are involved in non-oxidative functions, such as lipid synthesis [29] and biogenesis of organelle membranes, at least one isoform, ACSL-5, is believed to be associated with the mitochondrial outer membrane, suggesting a role for partitioning long-chain fatty acids toward oxidation [20,22,23,30,31,32].

Despite the popularity of the aforementioned hypothesis linking reduced mitochondrial lipid oxidation with disruptions in the insulin signaling cascade, more recent studies also suggest a relationship between mitochondrial fatty acid uptake and production of free radicals. More specifically, studies from our laboratory have recently linked lipid induced mitochondrial free radical production with reductions in insulin sensitivity [5].

Skeletal muscle is responsible for the majority of lipid oxidation in mammals; however, the role of ACSL-5 in regulating skeletal muscle mitochondrial fatty acid oxidation and the potential indirect influence over mitochondrial electron transport chain function and insulin action in humans remains unknown. Therefore, the purpose of this study was to investigate the potential role of ACSL-5 in mitochondrial function, including lipid oxidation, free radical formation, and insulin signaling. Herein, we report the novel findings that ACSL-5 in human skeletal muscle functions to increase fatty acid oxidation in mitochondria, in conjunction with unexpected findings of increased free radical production and reduced insulin signaling.

## 2. Materials and Methods 

### 2.1. Human Subjects and Tissue Biopsy

Six premenopausal, non-diabetic obese women participated in this study. Obese subjects (body mass index (BMI) = 34.7 ± 3.1 kg/m^2^; ages 26–44) were chosen because of reported reductions in skeletal muscle fatty acid oxidation capacity [7,10]. We specifically targeted African women (at least second generation verified by questionnaire and interviews during subject consent) because of compromised ACSL activity [18,19]. Prescreening by questionnaires and personal interviews demonstrated that subjects were: (1) sedentary (less than 30 min of physical activity per week for >6 months), (2) non-smokers, and (3) had no history of metabolic disease (no subjects were taking medications known to alter metabolism such as a thyroid hormone replacement). Skeletal muscle biopsies were obtained from the lateral aspect of the vastus lateralis using a modification of the percutaneous needle biopsy technique as previously reported [33]. Experiments were approved by the Institutional Review Board of East Carolina University.

### 2.2. Blood Sampling and Determination of Insulin Sensitivity 

Subjects reported to the laboratory in the fasted (10 h) state. A catheter was placed in the antecubital vein and the subjects rested for 30 min prior to blood sampling. Plasma was collected in ice cold ethylenediaminetetraacetic acid (EDTA) treated tubes. Plasma from fasted subjects was analyzed for glucose (YSI 2300 STAT Plus Glucose and Lactate Analyzer, YSI Inc; Yellow Springs, OH, USA) and insulin (Access Immunoassay System, Beckman-Coulter; Fullerton, CA, USA). A homeostasis model assessment value for insulin resistance (HOMA-IR) was calculated (HOMA-IR = (glucose, mg·dL^−1^ × insulin, μU·mL^−1^)/405). Insulin sensitivity was defined according to Stern et al. [34]: insulin sensitive, HOMA-IR < 3.60; insulin resistant, HOMA-IR > 3.60.

### 2.3. Cell Culture and Transfection

As described previously [35], skeletal muscle fibers (60–80 mg) were transferred to chilled Dulbecco’s Modified Eagles Medium (DMEM), and all visible fat and connective tissue were removed before cell culture. Cells were suspended in growth media containing DMEM supplemented with 10% fetal bovine serum (FBS), 0.5 mg/mL bovine serum albumin (BSA), 0.5 mg/mL fetuin, 20 ng/mL human epidermal growth factor, 0.39 μg/mL dexamethasone, and 50 μg/ml gentamicin/amphotericin B, and cultured at 37 °C in a humidified 5% CO_2_ and ambient air incubator. After reaching 70–80% confluence, myoblasts were harvested via trypsinization and then counted to determine cell density. The myoblasts (10^6^ cells) were resuspended in 100 μL Nucleofector solution (Lonza, Walkersville, MD, USA) combined with 3 μg ACSL-5 plasmid DNA (Invitrogen: Carlsbad, CA, USA) and 2 μg pmaxGFP (green fluorescent protein) vector (Lonza: Walkersville, MD, USA) for the experimental group, or only 2 μg pmaxGFP vector without ACSL-5 plasmid DNA for the control group. The control cells were obtained from the same subjects as the experimental group. The cells/DNA suspension was transfected using Amaxa’s Nucleofector Technology (Lonza: Walkersville, MD, USA) according to the manufacturer’s protocol. After transfection, the myoblasts were transferred to pre-equilibrated growth media plates for 48 h to reach 80% confluence. Growth media was switched to differentiation media containing 2% horse-serum, 0.5 mg/mL BSA, 0.5 mg/mL fetuin, and 50 µg/mL gentamicin/amphotericin B for the differentiation of myoblasts into myotubes. The transfection efficiency of green fluorescent protein (GFP) and the cell viability were measured by inverted fluorescence microscopy. Primary human myotubes were harvested and assayed on day 7 of differentiation for all cellular experiments as previously reported by our group [35]. Verification of transfection efficiency was confirmed via Western blot analysis using an ACSL-5 antibody (1:1000, Santa Cruz Biotechnology: Santa Cruz, CA, USA).

### 2.4. Measurement of Fatty Acid Oxidation 

Experiments utilizing [1-^14^C] palmitate was performed to measure fatty acid oxidation in the primary human skeletal myotubes according to modified methods of Cortright et al. [19]. Measurements for fatty acid oxidation in primary human myotubes have been previously validated by our group [36]. In brief, differentiated human skeletal muscle cells with or without ACSL-5 transfection were incubated in sealed reaction plates at 37 °C in a humidified 5% CO_2_ and ambient air incubator for 3 h. Palmitate oxidation was performed in differentiation media containing 100 µM palmitate, 12.5 mM 4-(2-hydroxyethyl)-1-piperazineethanesulfonic acid (HEPES), 0.25% BSA, 1mM carnitine, 5 mM glucose, and 10 µCi/mL [1-^14^C] palmitate (Sigma-Aldrich, St. Louis, MO, USA). Following the incubation period, the reactions were terminated via the addition of 50 µL 70% perchloric acid. The incubation plate was transferred to an orbital shaker, and ^14^CO_2_ was trapped in the adjoining well (200 µL of 1 N NaOH) for 1 h. Radioactivity as ^14^CO_2_ (complete palmitate oxidation) and ^14^C-ASM (acid soluble metabolites; incomplete palmitate oxidation) was determined via liquid scintillation counting using 4 mL of Uniscint BD (National Diagnostics, Atlanta, GA). The remaining cell pellets were washed twice with ice-cold phosphate-buffered saline, harvested in 200 µL 0.05% sodium dodecyl sulfate (SDS), and stored at −80 °C for subsequent protein determination. 

### 2.5. Preparation of Permeabilized Cells

After harvesting differentiated myotubes and cell counting, samples were centrifuged at 1000 rpm for 5 min and treated with 3 µg/10^6^ cells/mL digitonin—a mild, cholesterol-specific detergent—at 37 °C for 5 min with rotation. Digitonin selectively permeabilizes the sarcolemmal membranes while keeping mitochondrial membranes intact. Following permeabilization, cells were washed using centrifugation at 1000 rpm for 5 min to remove endogenous substrates. Permeabilized cells were used for mitochondrial O_2_ respiration as previously described [35].

### 2.6. Measurement of Mitochondrial O_2_ Respiration

As described previously by our group [35], high resolution oxygen (O_2_) consumption measurements were conducted at 37 °C using the Oroboros O_2_K Oxygraph (Innsbruck, Austria) in MiR05 respiration buffer containing 130 mM sucrose, 60 mM C_6_H_11_O_7_K, 1mM ethylene glycerol-bis(β-aminoethyl ether)-*N*,*N*,*N′,N′*-tetraacetic acid (EGTA), 3 mM MgCl_2_, 10 mm K_2_HPO_4_, 20 mM HEPES, and 1 mg/mL BSA (pH 7.4). After permeabilization, cells (1.0 × 10^6^ cells per chamber) were placed into separate chambers with 2 mL of MiR05. Mitochondrial O_2_ consumption was measured as follows: (i) 2 mM ADP/0.5 mM ATP (state 3 respiration following addition of substrate), (ii) 100 µM palmitate + 0.1 mM CoA + 1mM carnitine (fatty acid substrates), (iii) 100 µM malonyl-CoA (CPT-1 inhibitor), (iv) 2 mM glutamate + 1mM malate (complex I substrates), (v) 3 mM succinate (complex II substrate), (vi) 10 μg/ml oligomycin (inhibitor of mitochondrial ATP synthase, state 4 respiration), and (vii) 2 μM carbonylcyanide-p-trifluoromethoxyphenylhydrazone (FCCP, a protonophoric uncoupler; indicator of maximal respiratory capacity). Mitochondrial integrity was confirmed by addition of cytochrome C, while absence of endogenous substrates was confirmed by no increase in O_2_ consumption with the addition of ADP prior to substrate additions.

### 2.7. Measurement of Mitochondrial O_2_^−^ production 

Mitochondrial superoxide (O_2_^−^) was measured with flow cytometry (FACS-calibur; BD Bioscience, San Jose, CA, USA) in both control (only GFP transfected) and experimental cells (ACSL-5/GFP plasmid DNA transfected) after staining with mitochondrial superoxide specific dye MitoSOX Red (Molecular Probes, Eugene, OR, USA) as previously reported [37]. Briefly, on day 7 of differentiation (post transfection), 5 μM MitoSOX was applied for 15 min at 37 °C in a CO_2_ incubator protected from light followed by washing two times with Hank’s Buffered Salt Solution (HBSS). After trypsinization and neutralization with FBS, cells were counted and centrifuged at 1000 rpm for 10 min and resuspended with FACS buffer (PBS containing 1% BSA, 0.01% NaN_3_). For the positive control, the cells were treated with 50 µM Antimycin A for 60 min. MitoSOX Red was excited using laser exposure at 488 nm and the data were collected using FSC (forward scatter), SSC (side scatter), FL2 (588/42 nm), and FL3 (670LP) channels. In this study, the data were presented from the FL2 channel. Quantifications were performed from the mean intensity of MitoSOX fluorescence from triplicates.

### 2.8. Western Immunoblot Analysis for Insulin Signaling

Protein levels for phospho-/total IRS-1, phospho-/total Akt, phospho-/total AS160, and MnSOD were determined in the cytoplasmic protein fraction via Western immunoblot analysis. On day 7 of differentiation, the myotubes were treated with 100 nM porcine insulin (Sigma-Aldrich, St. Louis, MO, USA) for 10 min and then harvested in lysis buffer containing 50 mM HEPES, 10mM EDTA, 100 mM NaF, 50 mM Na pyrophosphate, 10 mM Na orthovanadate, and 1% Triton X-100 supplemented with phosphatase and protease inhibitor cocktails (Sigma-Aldrich, St. Louis, MO, USA). The protein concentration was determined using a BCA assay kit (Pierce Biotechnology, Rockford, IL, USA). Forty micrograms of protein from myotube homogenates in sample buffer (+5% β-mercaptoethanol) were then loaded into the wells of 12.5 % polyacrylamide gels, and electrophoresed at 150 V. Proteins were then transferred at 100 V for 2 h onto a polyvinylidene fluoride (PVDF) membrane. After staining with Ponceau S (Sigma-Aldrich, St. Louis, MO, USA) to verify equal loading, the membranes were blocked in 5% nonfat milk in tris-buffered saline (TBS) with 0.1% Tween-20 for 4 h. After blocking, membranes were incubated at 4 °C in blocking buffer for 12 h with the appropriate primary antibodies: rabbit polyclonal anti-phospho-IRS-1 (1:1000, Upstate: Lake Placid, NY, USA), rabbit polyclonal anti-IRS-1 (1:1000, Upstate: Lake Placid, NY, USA), rabbit polyclonal anti-phospho-Akt1/2/3 (Ser 473) (1:500, Santa Cruz Biotechnology: Santa Cruz, CA, USA), rabbit polyclonal anti-Akt1/2/3 (H-136) (1:500, Santa Cruz Biotechnology: Santa Cruz, CA, USA), rabbit polyclonal anti-phospho-AS160 (T642) (1:1000, Invitrogen: Carlsbad, CA, USA), and rabbit polyclonal anti-AS160 (1:4000, Upstate: Lake Placid, NY, USA), and rabbit polyclonal anti-MnSOD (1:10,000, Stressgen: Ann Arbor, MI, USA). Following three washings in TBS with 0.1% Tween-20, membranes were incubated at room temperature for 60 min in blocking buffer with horseradish peroxidase (HRP)–conjugated secondary antibodies (Santa Cruz Biotechnology). An enhanced chemiluminescence (ECL) detection system (Amersham: Piscataway, NJ, USA) was used for visualization. The membranes were stripped and re-probed with a GAPDH antibody (1:4000, Advanced Immunochemical: Long Beach, CA, USA) to verify equal loading among lanes as an internal control. Densitometry (as area times grayscale relative to background) was performed using a Kodak film cartridge and film, a scanner interfaced with a microcomputer, and the NIH Image Analysis 1.62 software program (National Institutes of Health, Bethesda, MD, USA).

### 2.9. Morphological Imaging 

Morphological changes and survival of transfected cells were monitored by obtaining photomicrographs under an inverted phase contrast microscopy (Olympus America Inc.: Melville, NY, USA) with a digital camera.

### 2.10. Statistics 

All experiments were conducted in parallel with experimental and control cells from the same subjects. For all data analysis except group comparisons for insulin signaling, a Student’s paired *t*-test was employed to determine the existence of mean differences between the ACSL-5 transfected experimental group and control group. For insulin signaling data, group means were compared using a one way ANOVA. Data were presented means ± standard error mean (SEM), with statistical significance between groups established a priori at *p* < 0.05.

## 3. Results

### 3.1. Human Subjects

Table 1 presents the characteristics for the subjects. The mean value for BMI was approximately 35 kg/m^2^, thus indicating that they were not morbidly obese. Participants were not considered insulin resistant as determined using HOMA-IR (2.8 ± 0.4).

### 3.2. GFP Fluorescence and ACSL-5 Overexpression via Transfection 

To determine transfection efficiency using the Amaxa Nucleofector device, primary human skeletal myoblasts were transfected via positive control vector GFP plasmid DNA. Cell viability was approximately 40% and transfection efficiency was approximately 50% at 24 h after transfection in myoblasts (Figure 1A). Seven days of differentiation demonstrated a strong expression of GFP fluorescence with fully differentiated cells being elongated and multinucleated (Figure 1B). These data indicate successful transfection using GFP plasmid DNA in differentiated primary human skeletal muscle cells.

Given the above evidence of transfection using GFP plasmid DNA, we determined the effect of ACSL-5 overexpression on the protein levels of ACSL-5. The protein levels of ACSL-5 were approximately 2-fold higher (*p* < 0.05) in ACSL-5 overexpressed cells compared with the control (Figure 1C). We also demonstrated that ACSL-5 overexpression did not affect differentiation based on the morphology of control (Figure 1D) versus ACSL-5 overexpression (Figure 1E).

### 3.3. ACSL-5 Overexpression Increased Complete and Incomplete Fatty Acid Oxidation 

To determine the effects of ACSL-5 overexpression on fatty acid oxidation in primary human skeletal myotubes, we measured radioactivity of CO_2_ (complete fatty acid oxidation) and ASM (incomplete fatty acid oxidation) using liquid scintillation counting. According to captured 1-^14^C-labeled CO_2_, oxidation of palmitate prepared from human skeletal myotubes was significantly elevated (+112%; *p* < 0.05) in ACSL-5 overexpressing cells versus control (Figure 2A).

In addition, ACSL-5 overexpression resulted in a significant increase (+71%; *p* < 0.05) in incomplete (^14^C-labeled ASM radioactivity) palmitate oxidation when compared with the control (Figure 2B). Our findings are consistent with the hypothesis that ACSL-5 overexpression leads to an increase of fatty acid oxidation in primary human skeletal myotubes. 

### 3.4. ACSL-5 Overexpression Increases Maximal Mitochondrial Respiratory Capacity 

Following activation of a long-chain fatty acid by ACSL, several steps are required for its complete oxidation. This includes transport (CPT-1) and matrix specific pathways including β-oxidation, the tricarboxylic acid (TCA) cycle, and oxidation of reducing equivalents by the electron transport chain to support ATP synthesis. Given the above evidence that ACSL-5 overexpression led to a significant increase in maximal fatty acid oxidation, we wished to examine the potential impact of its overexpression on the mitochondrial respiratory system. Using digitonin-permeabilized skeletal myotubes, maximal (ADP-stimulated, state 3) oxygen consumption rate was determined during respiration supported by palmitate followed by additional substrates added sequentially, and under maximal respiratory activity via addition of the uncoupler FCCP (Figure 3). O_2_ consumption supported by activated long-chain fatty acids (palmitate + CoA + carnitine) in the presence of ADP and ATP (state 3) were greater in ACSL-5 overexpressing cells but did not reach statistical significance (*p* = 0.14). To determine whether ACSL-5 overexpression influences the inhibitory effect of malonyl-CoA (CPT-1 inhibitor) on fatty acid oxidation, malonyl-CoA was added during ADP-stimulated respiration supported by long-chain fatty acids. However, there was no alteration in O_2_ consumption by the CPT-1 inhibitor between control and ACSL-5 overexpressing cells. In addition, ACSL-5 overexpression did not affect ADP-stimulated mitochondrial respiration (state 3) supported by both glutamate + malate (complex I substrates) and succinate (complex II substrate) in primary human skeletal myotubes. However, basal (non-ADP stimulated state 4) respiration by oligomycin and maximal uncoupled respiration by FCCP were greater (*p* < 0.05) in ACSL-5 overexpressed cells, indicative of greater fatty acid catabolic flux capacity following treatment. 

### 3.5. ACSL-5 Overexpression Reduces Insulin Siganling 

Elevations in fatty acyl-CoA are associated with lipid storage and insulin resistance through increased diacylglycerides, serine kinases, or ceramide production in skeletal muscle [16]. In order to determine whether ACSL-5 overexpression results in alterations in insulin signaling, we measured the protein levels of IRS-1, Akt, and AS160 in differentiated primary human skeletal myotubes.

Protein levels of p-IRS-1 (Ser307), total IRS-1 or p- to total IRS-1 ratio were not altered by ACSL-5 overexpression in human skeletal myotubes compared with non-ACSL-5 transfected control (Figure 4A). In addition, there were no effects of insulin treatment on protein levels in either control cells or ACSL-5 overexpressing cells (Figure 4A). However, the protein levels of p-Akt (Ser473) and p- to total Akt ratio in human skeletal myotubes were significantly lower (*p* < 0.05) compared with insulin-stimulated control cells (total Akt protein levels not affected) (Figure 4B). Moreover, insulin treatment resulted in a significant increase (*p* < 0.05) in protein levels of p-Akt only in control cells (Figure 4B). AS160, the downstream target of Akt that inhibits GLUT-4 translocation to the cell surface via inactivation of Rab GTPase-activating protein (GAP), expressed a pattern similar to Akt with ACSL-5 overexpression (Figure 4C). The protein levels of p-AS160 (T642) and p- to total AS160 ratio were decreased (*p* < 0.05) by ACSL-5 overexpression in primary human skeletal myotubes compared with control without affecting total AS160 protein levels (Figure 4C).

### 3.6. ACSL-5 Overexpression Increased Mitochondrial Oxidative Stress 

Oxidation of fatty acids is associated with elevated mitochondrial reactive oxygen species (ROS) production [38]. Mitochondrial ROS production has been linked to the development of insulin resistance in skeletal muscle [5]. To examine if ACSL-5 overexpression influences mitochondrial ROS production, mitochondrial superoxide production and manganese superoxide dismutase (MnSOD catalyzes dismutation of superoxide to H_2_O_2_) expression were determined in differentiated human skeletal muscle cells. Mean fluorescence intensity of oxidized MitoSOX Red determined using FACS analysis was increased (*p* < 0.05) by 30 % in primary human skeletal myotubes overexpressing ACSL-5 (Figure 5A–D).

Moreover, MnSOD protein was significantly higher (+119%; *p* < 0.05) in ACSL-5 overexpressing compared with control cells (Figure 5E), suggesting an adaptive response to elevated ROS by ACSL-5 overexpression. These findings demonstrate that ACSL-5 overexpression increased skeletal muscle mitochondrial ROS generation and that this may be associated with alterations of certain aspects of the insulin signaling cascade. 

## 4. Discussion

To our knowledge, this is the first report to demonstrate that elevated ACSL-5 expression in human skeletal muscle functions to increase mitochondrial fatty acid oxidation. However, contrary to conventional wisdom’s prediction that elevating mitochondrial fatty acid oxidation would attenuate the progression toward insulin resistance in skeletal muscle [3], ACSL-5 was also associated with increased free radical production (i.e., O_2_^−^ ) and reduced insulin signaling.

The present data demonstrate that ACSL-5 overexpression resulted in increased complete fatty acid oxidation (Figure 2A) and mitochondrial respiration in the absence of an inhibitory effect of malonyl-CoA (Figure 3), suggesting that ACSL-5 overexpression induced increases in fatty acid flux into the mitochondria. Our findings are consistent with previous studies in other tissues. For example, Zhou et al. [30] reported that overexpression of ACSL-5 in liver resulted in fatty acid partitioning toward β-oxidation.

Overexpression of ACSL-5 also increased the ASM production of palmitate (metabolites from β-oxidation and the TCA cycle) (Figure 2B). This is clinically important for cellular bioenergetics as Koves et al. [39] reported that insulin resistance was associated with increased incomplete fatty acid oxidation in skeletal muscle and argued for a connection between the development of insulin resistance and lipid-induced mitochondrial overload by incomplete fatty acid oxidation. The authors put forth the hypothesis that insulin resistance might result from excessive rather than reduced β-oxidation in skeletal muscle.

To extend the hypothesis by Koves et al. [39], we suggest that increased lipid supply to the mitochondria following ACSL-5 overexpression could lead to elevated NADH and FADH_2_ from β-oxidation, indicative of an imbalance between substrate availability and energetic demands. If β-oxidation is increased by oversupply of fatty acids without a corresponding increase in energy demand, as likely occurs in the obese human, surplus reducing equivalents are generated, which would result in an increased mitochondrial membrane potential, leading to increased mitochondrial oxidative stress [40,41]. This elevation in mitochondrial reactive oxygen species (ROS) generation could result in impaired insulin signaling in skeletal muscle [42,43]. Support for this contention can be found in the recent report by our group, which provides convincing evidence of a direct cause-and-effect relationship between increased mitochondrial lipid influx, excess ROS production, and attenuation of glucose uptake/insulin sensitivity [5]. Thus, we recently demonstrated that administration of a mitochondrial targeted antioxidant (SS31) and elevated mitochondrial-targeted catalase prevented high fat diet-induced insulin resistance [5], suggesting that mitochondrial ROS may be a critical factor in the etiology of insulin resistance in skeletal muscle. The current findings are consistent with this hypothesis as we found that mitochondrial superoxide (O_2_^•−^) was significantly increased by ACSL-5 overexpression in differentiated primary human skeletal muscle cells (Figure 5A–D). Moreover, MnSOD protein levels were dramatically increased (Figure 5E). One explanation for these observations is that the elevation in MnSOD represents a provoked cellular defense response to buffer elevating mitochondrial superoxide content/production during conditions of cellular lipid overload, but that this compensation was insufficient to completely curtail a rise in total ROS production. Together, ACSL-5 overexpression may induce insulin resistance through a mechanism of increased mitochondrial ROS production in skeletal muscle.

Given the above observations, we elected to examine the potential effects of ACSL-5 overexpression on the insulin signaling cascade. Overexpression led to an impairment of the insulin signaling cascade by reducing the phosphorylation of Akt and downstream AS160 in the absence of changes in upstream IRS-1 serine phosphorylation (Figure 4). In the obese population, there may be a hyper-serine phosphorylation of IRS-1 that masked any effects of ACSL-5 overexpression at that phosphorylation event, yet might trigger downstream signaling. Moreover, IRS-1 phosphorylation by serine/threonine kinases (e.g., IKKβ, JNK, PKCθ) may be associated with its impaired phosphorylation by tyrosine residues. Therefore, future directions could examine other phosphorylation events such as tyrosine. In this context, many studies linking obesity with reductions in insulin signaling have indicated that diminished mitochondrial function/content is the initial homeostatic alteration associated with the development of insulin resistance in skeletal muscle [6,11,44,45,46,47,48]. Thus, earlier attention had focused on accumulation of lipid intermediates such as acyl-CoA, diacylglycerol or ceramides that inhibit insulin signaling in skeletal muscle [3,6,15,16,17]. These findings provide support for the hypothesis that diminished fatty acid import and oxidation in mitochondria result in increased lipid intermediates, such as acyl-CoA, that are the provocateurs of reduced insulin signaling and insulin resistance. Our findings do not support these earlier observations entirely as an increased mitochondrial fatty acid oxidation or increased basal and maximal respiration (FCCP) were not associated with an improvement, but rather a decline in insulin signaling. Interestingly, we had earlier demonstrated that long-chain fatty acyl-CoA accumulation in skeletal muscle is not due solely to a reduction in fatty acid oxidation [10], nor is obesity itself always associated with reduced mitochondrial oxidation of fatty acids [49,50,51]. Thus, other reports and the present study suggest at least a partial disconnection between reduced mitochondrial oxidation of fatty acids and reductions in skeletal muscle insulin signaling and resultant insulin resistance reported with obesity [52].

A limitation of the current study may be the investigation of ACSL-5 overexpression in skeletal muscle cells solely from obese African-American women. However, we chose this approach intentionally because obese African-American women are reported to demonstrate attenuation of ACSL activity in skeletal muscle [18,19] and because of our reports of reduced fatty acid oxidation capacity in obese subjects [7,10]. Thus, it was not the intent of the current studies to make comparisons among racial groups, between males and females, nor to compare lean versus obese subjects. Although such comparisons are warranted in future studies, we conjectured in the current study that ACSL-5 overexpression would restore the reduced oxidative capacity of obese African-American women and accordingly found that ACSL-5 overexpression increased fatty acid oxidation in human skeletal muscle. However, the results also show that enhanced ACSL-5-dependent fatty acid oxidation occurred under conditions of concomitant increases in mitochondrial oxidative stress. The hypothesis has been put forth that inductions of fatty acid oxidation would serve to increase the supply of reducing molecules (i.e., NADH and FADH_2_), which would increase electron “slippage” from the electron transfer chain (ETC) and increase superoxide production and H_2_O_2_ production [5]. In addition, this study only focused on the effects of ACSL-5 overexpression on fatty acid oxidation, ROS, and insulin signaling in human skeletal muscle. However, it is warranted in future studies to examine the effects of ACSL-5 gene knockdown as well. 

A second limitation to the current study may lie with the experimental condition to test the potential changes in ACSL-5 expression on lipid metabolism in cell culture. Although the approach has been acceptable in the published literature from our and other laboratories, we acknowledge the potential limitation of the model when predicting influences of ACSL-5 in vivo with respect to PO_2_ levels in vitro versus in vivo cellular conditions (e.g., the current study utilized oxygen tensions of 20% vs. ≈4% in vivo). To this end, future studies in transgenic animal models overexpressing ACSL-5 are needed.

Finally, a potential limitation for the current study may lie with superoxide measurements using mitoSOX. Given that there is an increase in MnSOD content with ACSL-5 overexpression, it may be that the mitoSOX probe can out-compete the binding of MnSOD to superoxide. Thus, measurement of superoxide may be underestimated. Further studies are needed to confirm the understanding of ACSL-5 overexpression on ROS production. As such, further studies using high resolution oximetry and measurements of H_2_O_2_ production are warranted as published earlier by our laboratory [53]. 

## 5. Conclusions

In conclusion, this is the first report of a specific ACSL isoform (ACSL-5) influencing the mitochondrial oxidation of fatty acids in human skeletal muscle. Given contrasting findings from previous reports in hepatocytes, it is likely that this isoform can function in a diverse manner depending on tissue type. ACSL-5 overexpression resulted in increased complete fatty acid oxidation and basal/maximal uncoupled respiration in differentiated primary human skeletal muscle cells. However, ACSL-5 overexpression was also associated with increases in incomplete fatty acid oxidation and mitochondrial oxidative stress (e.g., elevations in O_2_^−^) which might, by mechanism(s) yet to be discovered, result in impairment of insulin signaling suggesting a coupling between cellular lipid supply and energy demand, shifts in mitochondrial membrane potential, cellular REDOX state, and insulin resistance in human skeletal muscle. These findings also demonstrate that in human skeletal muscle, ACSL-5 is a contributor to mitochondrial lipid partitioning, thereby extending past observations in rodent skeletal muscle suggesting that ACSL-1 and perhaps FATP-1 are involved in the mitochondrial partitioning of lipids [28,54]. 

## Figures and Tables

**Figure 1 ijerph-16-01157-f001:**
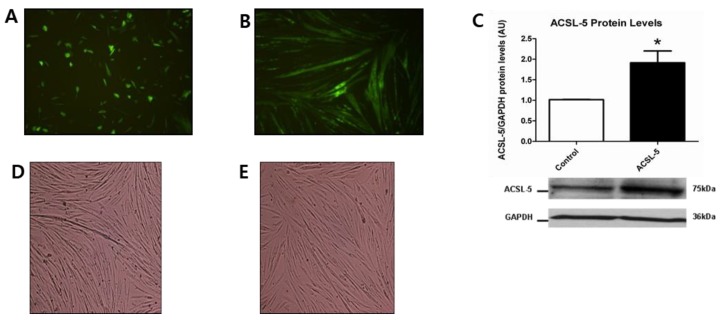
GFP fluorescence and ACSL-5 overexpression with transfection in primary human skeletal muscle cells (10X magnification). (**A**) Primary human skeletal myoblasts were transfected using positive control vector GFP without ACSL-5 plasmid DNA to verify transfection. Results indicated an approximately 50% transfection efficiency at 24 h after transfection. (**B**) At day 7 of the differentiation period, GFP fluorescence was strongly expressed in the primary human skeletal muscle cells. (**C**) Transfection efficiency of ACSL-5 normalized to GAPDH (open bar, control cells; closed bar, ACSL-5 transfected cells). The protein levels of ACSL-5 were approximately 2-fold higher in ACSL-5 overexpressed cells than in control. Data are presented as mean ± SEM (n = 6). * *p* < 0.05 versus control. Blots were probed with antibodies recognizing ACSL-5 or GAPDH. (**D**,**E**) Effect of transfection on cell morphology (10× magnification). ACSL-5 overexpression did not affect differentiation in human skeletal muscle cells based on the morphology of control (D) and ACSL-5 overexpressed cells (E).

**Figure 2 ijerph-16-01157-f002:**
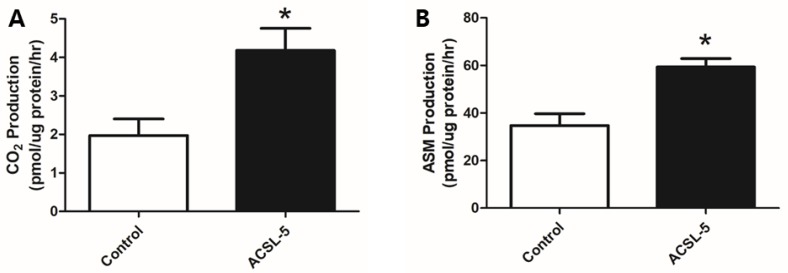
Effect of ACSL-5 overexpression on fatty acid oxidation. Fatty acid oxidation was measured using ^14^C-labeled radioactivity of CO_2_ (**A**) and ASM (**B**) via liquid scintillation counting (open bar, control cells; closed bar, ACSL-5 transfected cells). Palmitate complete oxidation (CO_2_ producton) was approximately 2-fold higher in the ACSL-5 transfected group compared with the GFP-only transfected control (A). Similarly, palmitate incomplete oxidation (ASM production) was significantly increased by ACSL-5 overexpression (B). Data are expressed as presented as mean ± SEM (n = 6). * *p* < 0.05 versus control.

**Figure 3 ijerph-16-01157-f003:**
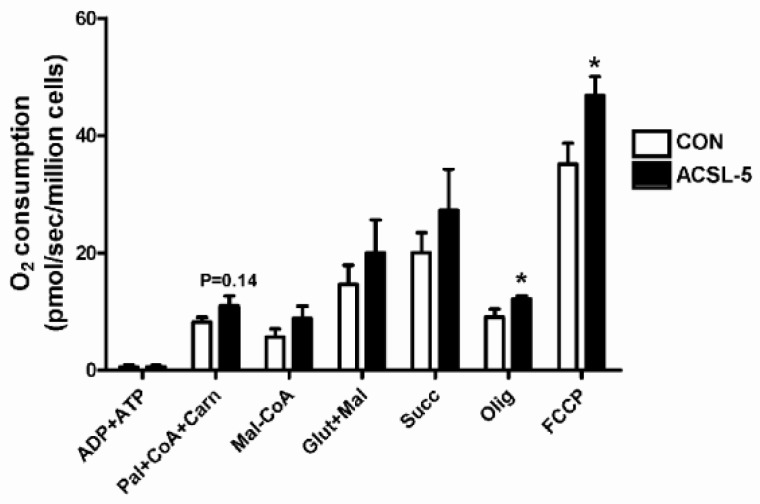
Effect of ACSL-5 overexpression on mitochondrial oxygen consumption measured in permeabilized primary human skeletal myotubes. The following multiple substrate protocols were commenced for mitochondrial O_2_ consumption: (i) 2 mM ADP/0.5 mM ATP (state 3 following addition of substrates), (ii) 0.1 mM palmitate + 0.1 mM CoA + 1 mM carnitine (fatty acid substrates, state 3), (iii) 0.1 mM malonyl-CoA (CPT-1 inhibitor, state 3), (iv) 2 mM glutamate + 1 mM malate (complex I substrates, state 3), (v) 3 mM succinate (complex II substrate, state 3), (vi) 10 μg/mL oligomycin (inhibitor of mitochondrial ATP synthase, non-ADP stimulated basal state 4), and (vii) 2 μM carbonylcyanide-p-trifluoromethoxyphenylhydrazone (FCCP, a protonophoric uncoupler, maximal uncoupled respiration) (open bar, control cells; closed bar, ACSL-5 transfected cells). The rate of O_2_ consumption was expressed as pmol/sec/million cells. Data are presented as mean ± SEM (n = 6). * *p* < 0.05 versus control (CON).

**Figure 4 ijerph-16-01157-f004:**
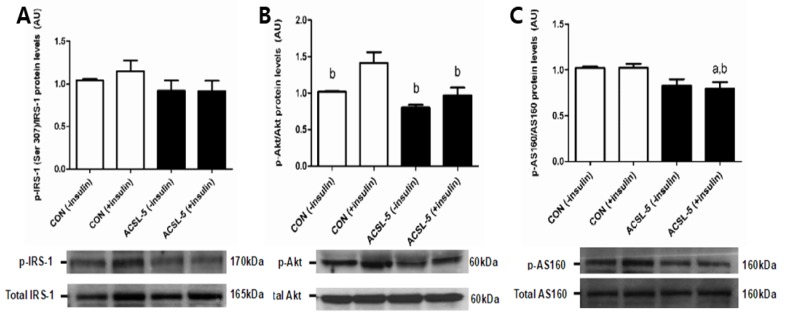
Effect of ACSL-5 overexpression on insulin signaling. Protein levels of phospho-/total IRS-1 (**A**), phospho-/total Akt (**B**), and phospho-/total AS160 (**C**) were measured using Western immunoblot analysis (open bar, control cells; closed bar, ACSL-5 transfected cells). Western blots and mean (±SEM) data (n = 6) in the absence or presence of insulin treatment were presented. ^a^
*p* < 0.05 versus non-insulin-treated control; ^b^
*p* < 0.05 versus insulin-treated control (CON).

**Figure 5 ijerph-16-01157-f005:**
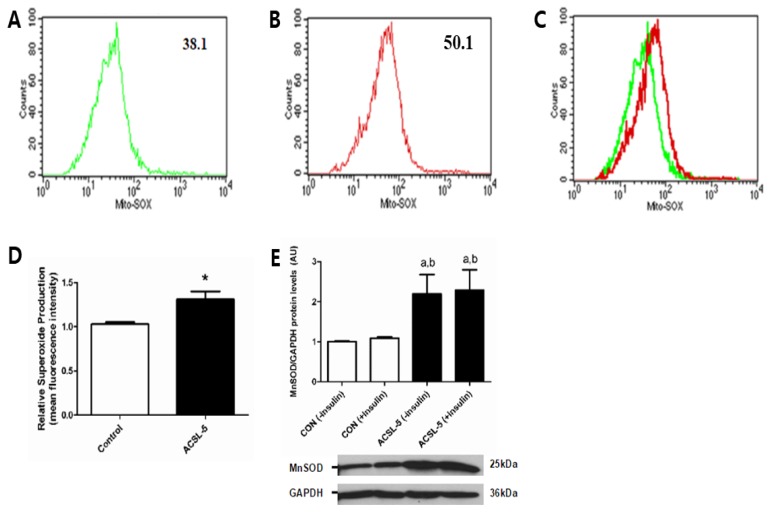
Effects of ACSL-5 overexpression on mitochondrial superoxide production (**A–D**) and MnSOD protein levels (**E**). Representative histograms of flow cytometry experiments were presented demonstrating mean fluorescence intensity (GFP-control: 38.1, ACSL-5 + GFP overexpression treatment: 50.1) of oxidized MitoSOX Red mitochondrial superoxide (O_2_^−^) indicator from control group (only GFP transfected cells) (A), ACSL-5 + GFP transfected group (B), and overlay with control and ACSL-5 overexpressed group (C). Relative superoxide production was significantly increased in ACSL-5 overexpressed group compared with the control (D). Data are presented as mean ± SEM (n = 5). * *p* < 0.05 versus control. The protein levels of MnSOD (anti-oxidant enzyme) were dramatically increased by ACSL-5 overexpression (open bar, control cells; closed bar, ACSL-5 transfected cells) (E). Western blots and mean (±SEM) data (n = 6) normalized to GAPDH in the absence or presence of insulin treatment were presented. **^a^**
*p* < 0.05 versus non-insulin-treated control; **^b^**
*p* < 0.05 versus insulin-treated control (CON).

**Table 1 ijerph-16-01157-t001:** Subject characteristics.

Variables	Obese Subjects
Numbers	6
Age (years)	33.0 ± 2.7
Height (cm)	167.2 ± 2.1
Weight (kg)	97.3 ± 8.3
BMI (kg/m^2^)	34.7 ± 3.1
% fat	44.9 ± 2.5
HOMA-IR	2.8 ± 0.4

Data from women recruited for skeletal muscle biopsies. Abbreviations: BMI, body mass index; HOMA-IR, homeostasis model assessment for insulin resistance.

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
