# Peer review of "Overexpression of Long-Chain Acyl-CoA Synthetase 5 Increases Fatty Acid Oxidation and Free Radical Formation While Attenuating Insulin Signaling in Primary Human Skeletal Myotubes"

_ijerph, 2019, doi:10.3390/ijerph16071157_

Round 1
Reviewer 1 Report
The manuscript by Kwak et al. is a well-designed and well-written study that seeks to understand the role of acyl CoA synthetase 5 (ACSL5) in human skeletal muscle fatty acid oxidation. Authors used to primary myoblast isolated from biopsy samples and overexpressed ACSL5 by transfection and assessed different parameters of muscle mitochondrial function.
This is an important study that enhances our understanding of long-chain acyl co A synthetase 5 in muscle fatty acid metabolism as well as in pathological metabolic disorders.
I don’t see any major issues and only have minor comments.
1. In the experimental design, ACSL5 overexpression was achieved in culture condition where myoblasts are grown in 20% O2 tension. Perhaps this is beyond the scope of the manuscript, but fatty acid oxidation in vivo is under ~ 4% O2 tension. Data observed in the study may not reflect in vivo condition.
2. Superoxide measurement with mitoSOX is not reliable. Given that there is an increase in MnSOD content with ACSL5 overexpression, it is hard to believe that mitoSOX probe can out-compete binding of MnSOD to superoxide. Thus, measurement of superoxide is most likely be underestimated.
Author Response
Thank you for your thorough reviews and constructive comments. We have attempted to fully address all of your concerns and comments within a revised manuscript. We feel that the suggested revisions have strengthened the manuscript. All changes in the manuscript are highlighted in red in the associated document. Specific responses to reviewer questions and comments are included below:
1. In the experimental design, ACSL5 overexpression was achieved in culture condition where myoblasts are grown in 20% O2 tension. Perhaps this is beyond the scope of the manuscript, but fatty acid oxidation in vivo is under ~ 4% O2 tension. Data observed in the study may not reflect in vivo condition.
Response to reviwer comment: Thank you for your insightful comment. As the reviewer has indicated, changes in environmental variables such as oxygen (20% in culture compared to 2-5% in vivo) can affect cell’s function and metabolic response. In this study, primary human skeletal muscle myoblasts were cultured at 37°C in a humidified 5% CO2 and ambient air incubator (20% oxygen tension) as typical cell culture conditions. We and other groups have published these methods in detail elsewhere (Huang et al., 2017; Consitt et al., 2010).
<References>
- Huang, T. Y.; Zheng, D.; Houmard, J. A.; Brault, J. J.; Hickner, R. C.; Cortright, R. N. Overexpression of PGC-1α increases peroxisomal activity and mitochondrial fatty acid oxidation in human primary myotubes. American Journal of Physiology. Endocrinology and Metabolism, 2017, 312(4), E253–E263.
- Consitt, L. A.; Bell, J. A.; Koves, T. R.; Muoio, D. M.; Hulver, M. W.' Haynie, K. R.; Dohm, G. L.;Houmard, J. A. Peroxisome proliferator-activated receptor-gamma coactivtor-1alpha overexpresson increases lipid oxidation in myocytes from extremely obese individuals. Diabetes, 2010, 59(6), 1407–1415.
However, we accept the reviewer’s comment and have included this as a second limitation to the present study as described in the discussion beginning with lines 445, pages 11-12.: “A second limitation to the current study may lie with the experimental condition to test the potential changes in ACSL-5 expression on lipid metabolism in cell culture. Although the approach has been acceptable in the published literature from our and other laboratories, we acknowledge the potential limitation of the model when predicting influences of ACSL-5 in vivo with respect to PO2 levels in vitro vs. in vivo cellular conditions (e.g., the current study utilized oxygen tensions of 20% vs. ~4% in vivo). To this end, future studies in transgenic animal models overexpressing ACSL 5 are needed.”
2. Superoxide measurement with mitoSOX is not reliable. Given that there is an increase in MnSOD content with ACSL5 overexpression, it is hard to believe that mitoSOX probe can out-compete binding of MnSOD to superoxide. Thus, measurement of superoxide is most likely be underestimated.
Response to reviwer comment: Thank you for this comment. We utilized MitoSOX Red (5uM) as it is specific to the mitochondria and permeates live cells rapidly. In fact, many studies (Johnson-Cadwell et al., 2007, Kwak et al., 2012) have used this to not only measure mitochondrial ROS, but to also localize subcellular ROS production. We found that MnSOD protein levels were dramatically increased by ACSL-5 overexpression. Also, we found that ACSL-5 overexpression increased mitochondrial superoxide as an indicator using MitoSOX Red and flow cytometry as previously reported. One explanation for these observations is that the elevation in MnSOD represents a provoked cellular defense response to buffer elevating mitochondrial superoxide content/production during conditions of cellular lipid overload. However, this compensation was insufficient to completely curtail a rise in total ROS production (lines 397-402, pages 10-11).
<References>
- Johnson-Cadwell, L. I.; Jekabsons, M. B.; Wang, A.; Polster, B. M.; Nicholls, D. G. “Mild Uncoupling” does not decrease mitochondrial superoxide levels in cultured cerebellar granule neurons but decreases spare respiratory capacity and increases toxicity to glutamate and oxidative stress. Journal of Neurochemistry, 2007, 101(6), 1619–1631.
- Kwak, H. B.; Thalacker-Mercer, A.; Anderson, E. J.; Lin, C. T.; Kane, D. A.; Lee, N. S.; Cortright, R. N.; Bamman, M. M.; Neufer, P. D. Simvastatin impairs ADP-stimulated respiration and increases mitochondrial oxidative stress in primary human skeletal myotubes. Free Radical Biology and Medicine, 2012, 52(1), 198-207.
We have described this potential limitation and suggest further studies in the discussion: “Finally, a potential limitation for the current study lies with superoxide measurements using mitoSOX. Given that there is an increase in MnSOD content with ACSL-5 overexpression, it may be that the mitoSOX probe can out-compete binding of MnSOD to superoxide. Thus, measurement of superoxide may be underestimated. Further studies are needed to confirm the understanding of ACSL-5 overexpression on ROS production. As such, further studies using high resolution oximetry and measurements of H2O2 production are warranted as published earlier by our laboratory.” (lines 452-458, page 12)

Reviewer 2 Report
The manuscript by Kwak et al. entitled “Overexpression of Long-Chain Acyl-CoA Synthetase 5 Increases Fatty Acid Oxidation and Free Radical Formation While Attenuating Insulin Signaling in Primary Human Skeletal Myotubes” under consideration for publication International Journal of Environmental Research and Public Health examines the effect of ACSL-5 over-expression on isolated muscle metabolism and insulin signaling. The report demonstrates that ACSL-5 plays a role in the oxidation of lipids within skeletal muscle mitochondria, which intriguingly correlates with reduced insulin signaling. In general, the report is well-written and interesting. The following considerations come to mind.
First, the authors should provide a reference that shows that myotube structure is maintained during flow cytometry experiments following trypsinization (in this reviewer’s mind, the tripsinization process can disrupt the myotube structure).
Secondly, in Figure 2 and throughout the manuscript, the authors describe CO2 oxidation and ASM oxidation as indicators of complete or incomplete lipid oxidation, respectively. However “oxidation” seems technically inaccurate. In this reviewer’s mind, CO2 is eliminated during lipid metabolism (not oxidized), and fatty acids are “oxidized”. Thus perhaps “CO2 production” or palmitate oxidation is a better description for the manuscript, figure legend, and related figure axis. Similarly, ASM oxidation seems like a misnomer because if it were oxidized, ASM would not be present. Again, perhaps ASM production or partial lipid oxidation are better descriptors of what was measured.
Lastly, the authors comment on the limited sample from which primary muscle was collected, and mention that comparisons between overweight and normal weight are beyond the scope of this manuscript (although are necessary topics for future research). However, given the authors also describe how the tested demographic (obese African-American subjects) exhibits reduced fatty acid oxidation capacity and reduced ACSL activity, it seems conceivable that the elevation of ACSL levels used during the experiments simply restores/rescues the reduced oxidative capacity of this demographic. Therefore this reviewer believes the paper could benefit from an expansion of the limitations section.
Author Response
Thank you for your thorough reviews and constructive comments. We have attempted to fully address all of your concerns and comments within a revised manuscript. We feel that the suggested revisions have strengthened the manuscript. All changes in the manuscript are highlighted in red in the associated document. Specific responses to reviewer questions and comments are included below:
First, the authors should provide a reference that shows that myotube structure is maintained during flow cytometry experiments following trypsinization (in this reviewer’s mind, the tripsinization process can disrupt the myotube structure).
Response to reviwer comment: Thank you for this thoughtful comment. As the reviewer indicated, the myotube structure is disrupted by tripsinization process for flow cytometry experiments. However, although the myotube structure is disrupted during short experiment time, we would assume that ACSL-5 overexpression-associated myotube function and gene expression are maintained. We just minimized trypsin exposure to avoid killing the cells. Also, we just focused on the comparison between control and ACSL-5 overexpression in this study.
Secondly, in Figure 2 and throughout the manuscript, the authors describe CO2 oxidation and ASM oxidation as indicators of complete or incomplete lipid oxidation, respectively. However “oxidation” seems technically inaccurate. In this reviewer’s mind, CO2 is eliminated during lipid metabolism (not oxidized), and fatty acids are “oxidized”. Thus perhaps “CO2 production” or palmitate oxidation is a better description for the manuscript, figure legend, and related figure axis. Similarly, ASM oxidation seems like a misnomer because if it were oxidized, ASM would not be present. Again, perhaps ASM production or partial lipid oxidation are better descriptors of what was measured.
Response to reviwer comment: This suggestion is indeed more accurate. The terminology of CO2 oxidation and ASM oxidation is replaced with CO2 production and ASM production throughout the manuscript.
Lastly, the authors comment on the limited sample from which primary muscle was collected, and mention that comparisons between overweight and normal weight are beyond the scope of this manuscript (although are necessary topics for future research). However, given the authors also describe how the tested demographic (obese African-American subjects) exhibits reduced fatty acid oxidation capacity and reduced ACSL activity, it seems conceivable that the elevation of ACSL levels used during the experiments simply restores/rescues the reduced oxidative capacity of this demographic. Therefore this reviewer believes the paper could benefit from an expansion of the limitations section.
Response to reviewer comment: According to the reviewer’s comment, we have modified the text content in the limitation section of discussion as below (lines 433-441, page 11).
: “we conjectured in this study that ACSL-5 overexpression would restore the reduced oxidative capacity of obese African-American women and accordingly found that ACSL-5 overexpression increased fatty acid oxidation in human skeletal muscle. However, the results also showed that enhanced ACSL-5 dependent fatty acid oxidation occurred under conditions of concomitant increases in mitochondrial oxidative stress. The hypothesis has been put forth that inductions of fatty acid oxidation would serve to increase supply of reducing molecules (i.e., NADH and FADH2) which would increase electron “slippage” from the electron transfer chain (ETC) and increase superoxide production and H2O2 production (Anderson et al., 2009).”
As the added oxidative stress has been implicated in the induction of insulin resistance, our findings of reduced insulin signaling is congruent with the published literature. Interestingly, African-Americans have been reported to have reduced mitochondrial oxidative capacity vs. Caucasian counterparts. (Cortright et al., 2006). Accordingly, we agree with the reviewer that because this was not tested, the absence of the racial differences need future study and should be acknowledged as limitation in the present manuscript. We have expanded the limitations in the dicussion as the reviewer's suggestion (lines 427-433, page 11).
<References>
- Anderson, E. J.; Lustig, M. E.; Boyle, K. E.; Woodlief, T. L.; Kane, D. A.; Lin, C. T.; Price, J. W., 3rd; Kang, L.; Rabinovitch, P. S.; Szeto, H. H.; et al. Mitochondrial H2O2 emission and cellular redox state link excess fat intake to insulin resistance in both rodents and humans. Journal of Clinical Investigation, 2009, 119, 573-581.
- Cortright, R. N.; Sandhoff, K. M.; Basilio, J. L.; Berggren, J. R.; Hickner, R. C.; Hulver, M. W.; Dohm, G. L.; Houmard, J. A. Skeletal muscle fat oxidation is increased in African-American and white women after 10 days of endurance exercise training. Obesity, 2006, 14, 1201-1210.

Reviewer 3 Report
The aim of this study was to assess the effect of long-chain fatty acid synthase-5 (ACSL-5) in human skeletal muscle cells. Skeletal myoblasts were isolated from vastus lateralis of obese women and transfected with ACSL-5 containing or blank vector. Overexpression of ACSL-5 increased complete and incomplete fatty acid oxidation, basal and maximal uncoupled respiration and mitochondrial superoxide production while reduced insulin activity (insulin-induced Akt and AS160 protein phosphorylation). The results suggest that enhanced ACSL-5 dependent fatty acid oxidation impairs skeletal muscle insulin signaling by stimulating mitochondrial oxidative stress.
The topic is of interest. The methods used were appropriate and the manuscript is overall well-written. However, there are some concerns/limitations regarding data interpretation.
1) The main approach was to increase ACSL-5 above physiological level. Supraphysiological ACSL-5 could have effects different than normal level of this protein. It would be of interest to examine the effect of ACSL-5 gene knockdown as well.
2) The study was performed in obese females. The effect of ACSL-5 on insulin sensitivity in males as well as in non-obese females may be different from that reported in this study.
3) How could be explained the finding that ACSL-5 overexpression had no effect on IRS-1 phosphorylation but reduced insulin-induced Akt phosphorylation? IRS-1 phosphorylation by serine-threonine protein kinases is often associated with its impaired phosphorylation at tyrosine residues induced by insulin. What mechanism is suggested to explain ACSL-5 induced insulin resistance? It would be of interest to measure insulin-induced IRS-1 phosphorylation at tyrosine residues to address this question.
Author Response
Thank you for your thorough reviews and constructive comments. We have attempted to fully address all of your concerns and comments within a revised manuscript. We feel that the suggested revisions have strengthened the manuscript. All changes in the manuscript are highlighted in red in the associated document. Specific responses to reviewer questions and comments are included below:
The main approach was to increase ACSL-5 above physiological level. Supraphysiological ACSL-5 could have effects different than normal level of this protein. It would be of interest to examine the effect of ACSL-5 gene knockdown as well.
Response to reviwer comment: Thank you for this suggestion. We will plan to do this in future experiments. Accordingly, we have added this future direction in the limitations section (lines 441-444, page 11). For reference, we found in the current study that there was only a doubling of ACSL-5 protein levels by ACSL-5 overexpression (Fig. 1C), which seems to be at least quasi physiological. Also, we found in the current study that ACSL-5 overexpression did not affect differentiation in human skeletal muscle cells based on the morphology of control and ACSL-5 overexpressed cells (Fig. 1D, 1E).
The study was performed in obese females. The effect of ACSL-5 on insulin sensitivity in males as well as in non-obese females may be different from that reported in this study.
Response to reviewer comment: We agree with the reviewer's comment. We plan to examine males vs. females as well as lean vs. obese subjects in the future studies. However, our current study was based on the findings in our group's previous study (Privette et al., 2003) as well as within the data that the prevalence of obesity is higher in African-American women. We have mentioned this limitation in the discussion (lines 431-433, page 11)
- Privette, J. D.; Hickner, R. C.; Macdonald, K. G.; Pories, W. J.; Barakat, H. A. Fatty acid oxidation by skeletal muscle homogenates from morbidly obese black and white American women. Metabolism, 2003, 52, 735-738.
How could be explained the finding that ACSL-5 overexpression had no effect on IRS-1 phosphorylation but reduced insulin-induced Akt phosphorylation? IRS-1 phosphorylation by serine-threonine protein kinases is often associated with its impaired phosphorylation at tyrosine residues induced by insulin. What mechanism is suggested to explain ACSL-5 induced insulin resistance? It would be of interest to measure insulin-induced IRS-1 phosphorylation at tyrosine residues to address this question.
Response to reviwer comment: A good point. In the obese population, there may be a hyper-serine phosphorylation of IRS-1 that masked any effects of ACSL-5 overexpression at that phosphorylation event, yet might trigger downstream signaling. This finding is consistent with other studies (Bikman et al., 2008) by which the authors hypothesize that skeletal muscle of obese subjects may have a so described endogenous brake on insulin signaling. However, future directions could examine other phosphorylation events such as tyrosine. We have added discussion of this possibilty in the discussion as per the reviewer comment (lines 406-411, page 11).
The potential mechanisms to explain ACSL-5 overexpression-induced insulin resistance may include the following. As we have stated in the discussion (lines 361-367, page 9), ACSL-5 overexpression increased the ASM production of palmitate (metabolites from β-oxidation and the TCA cycle) in the present study. This is a potential mechanism as Koves et al. (2008) reported that insulin resistance was associated with increased incomplete fatty acid oxidation in skeletal muscle and argued for a connection between the development of insulin resistance and lipid-induced mitochondrial overload by incomplete fatty acid oxidation. Second, mitochondrial ROS production by ACSL-5 overexpression may be linked to the development of insulin resistance in skeletal muscle (Anderson et al., 2009). This elevation in mitochondrial reactive oxygen species (ROS) generation could result in impaired insulin signaling in skeletal muscle. The above-mentioned answers are included in the manuscript (lines 381-388, page 10).
<References>
- Bikman, B. T.; Zheng, D.; Pories, W. J.; Chapman, W.; Pender, J. R.; Bowden, R. C.; Reed, M. A.; Cortright, R. N.; Tapscott, E. B.; Houmard, J. A.; et al. Mechanism for improved insulin sensitivity after gastric bypass surgery. Journal of Clincal Endocrinology and Metabolsim, 2008, 93(12), 4656-63.
- Koves, T. R.; Ussher, J. R.; Noland, R. C.; Slentz, D.; Mosedale, M.; Ilkayeva, O.; Bain, J.; Stevens, R.; Dyck, J. R.; Newgard, C. B.; et al. Mitochondrial overload and incomplete fatty acid oxidation contribute to skeletal muscle insulin resistance. Cell Metabolism, 2008, 7, 45-56.
- Anderson, E. J.; Lustig, M. E.; Boyle, K. E.; Woodlief, T. L.; Kane, D. A.; Lin, C. T.; Price, J. W., 3rd; Kang, L.; Rabinovitch, P. S.; Szeto, H. H.; et al. Mitochondrial H2O2 emission and cellular redox state link excess fat intake to insulin resistance in both rodents and humans. Journal of Clinical Investigation, 2009, 119, 573-581.
Round 2
Reviewer 3 Report
The manuscript has been revised according to the reviewers' comments. All concerns raised by the reviewers have been adequately addressed.